# Phases of cortical actomyosin dynamics coupled to the neuroblast polarity cycle

Chet Huan Oon, Kenneth E Prehoda*

Institute of Molecular Biology, Department of Chemistry and Biochemistry, University of Oregon, Eugene, United States

**Abstract** The Par complex dynamically polarizes to the apical cortex of asymmetrically dividing *Drosophila* neuroblasts where it directs fate determinant segregation. Previously, we showed that apically directed cortical movements that polarize the Par complex require F-actin (Oon and Prehoda, 2019). Here, we report the discovery of cortical actomyosin dynamics that begin in interphase when the Par complex is cytoplasmic but ultimately become tightly coupled to cortical Par dynamics. Interphase cortical actomyosin dynamics are unoriented and pulsatile but rapidly become sustained and apically-directed in early mitosis when the Par protein aPKC accumulates on the cortex. Apical actomyosin flows drive the coalescence of aPKC into an apical cap that depolarizes in anaphase when the flow reverses direction. Together with the previously characterized role of anaphase flows in specifying daughter cell size asymmetry, our results indicate that multiple phases of cortical actomyosin dynamics regulate asymmetric cell division.

## Editor's evaluation

Oon and Prehoda report pulsatile contraction of apical membrane in the process of Par protein polarization in Drosophila neuroblasts. This explains how/why actin filament was required to localize/ polarize Par complex. This very much resembles the observation in *C. elegans* embryos, and nicely unifies observations across systems.

*For correspondence:
prehoda@uoregon.edu

**Competing interest:** The authors declare that no competing interests exist.

## Introduction

The Par complex polarizes animal cells by excluding specific factors from the Par cortical domain (*Lang and Munro, 2017*; *Venkei and Yamashita, 2018*). In *Drosophila* neuroblasts, for example, the Par domain forms at the apical cortex during mitosis where it prevents the accumulation of neuronal fate determinants, effectively restricting them to the basal cortex. The resulting cortical domains are bisected by the cleavage furrow leading to fate determinant segregation into the basal daughter cell where they promote differentiation (*Homem and Knoblich, 2012*). It was recently discovered that apical Par polarization in the neuroblast is a multistep process in which the complex is initially targeted to the apical hemisphere early in mitosis where it forms a discontinuous meshwork (*Kono et al., 2019*; *Oon and Prehoda, 2019*). Cortical Par proteins then move along the cortex toward the apical pole, ultimately leading to formation of an apical cap that is maintained until shortly after anaphase onset (*Oon and Prehoda, 2019*). Here, we examine how the cortical movements that initiate and potentially maintain neuroblast Par polarity are generated.

An intact actin cytoskeleton is required for the movements that polarize Par proteins to the neuroblast apical cortex, but its role in the polarization process has been unclear. Depolymerization of F-actin causes apical aPKC to spread to the basal cortex (*Hannaford et al., 2018*; *Oon and Prehoda, 2019*), prevents aPKC coalescence, and induces disassembly of the apical aPKC cap (*Oon and Prehoda, 2019*), suggesting that actin filaments are important for both apical polarity initiation and

its maintenance. How the actin cytoskeleton participates in polarizing the Par complex in neuroblasts has been unclear, but actomyosin plays a central role in generating the anterior Par cortical domain in the *C. elegans* zygote. Contractions oriented toward the anterior pole transport the Par complex from an evenly distributed state (*Illukkumbura et al., 2020*; *Lang and Munro, 2017*). Bulk transport is mediated by advective flows generated by highly dynamic, transient actomyosin accumulations on the cell cortex (*Goehring et al., 2011*). While cortical movements of actomyosin drive formation of the Par domain in the worm zygote and F-actin is required for neuroblast apical Par polarity, no apically directed cortical actomyosin dynamics have been observed during the neuroblast polarization process, despite extensive examination (*Barros et al., 2003*; *Cabernard et al., 2010*; *Connell et al., 2011*; *Koe et al., 2018*; *Roth et al., 2015*; *Roubinet et al., 2017*; *Tsankova et al., 2017*). Instead, both F-actin and myosin II have been reported to be cytoplasmic or uniformly cortical in interphase, and apically enriched at metaphase (*Barros et al., 2003*; *Koe et al., 2018*; *Tsankova et al., 2017*), before undergoing cortical flows toward the cleavage furrow that are important for cell size asymmetry (*Cabernard et al., 2010*; *Connell et al., 2011*; *Roubinet et al., 2017*).

The current model for neuroblast actomyosin dynamics is primarily based on the analysis of fixed cells or by imaging a small number of medial sections in live imaging experiments and we recently found that rapid imaging of the full neuroblast volume can reveal dynamic phases of movement that are not detected with other methods (*LaFoya and Prehoda, 2021*; *Oon and Prehoda, 2019*). Here, we use rapid full volume imaging to investigate whether cortical actomyosin dynamics are present in neuroblasts when the Par complex undergoes its polarity cycle.

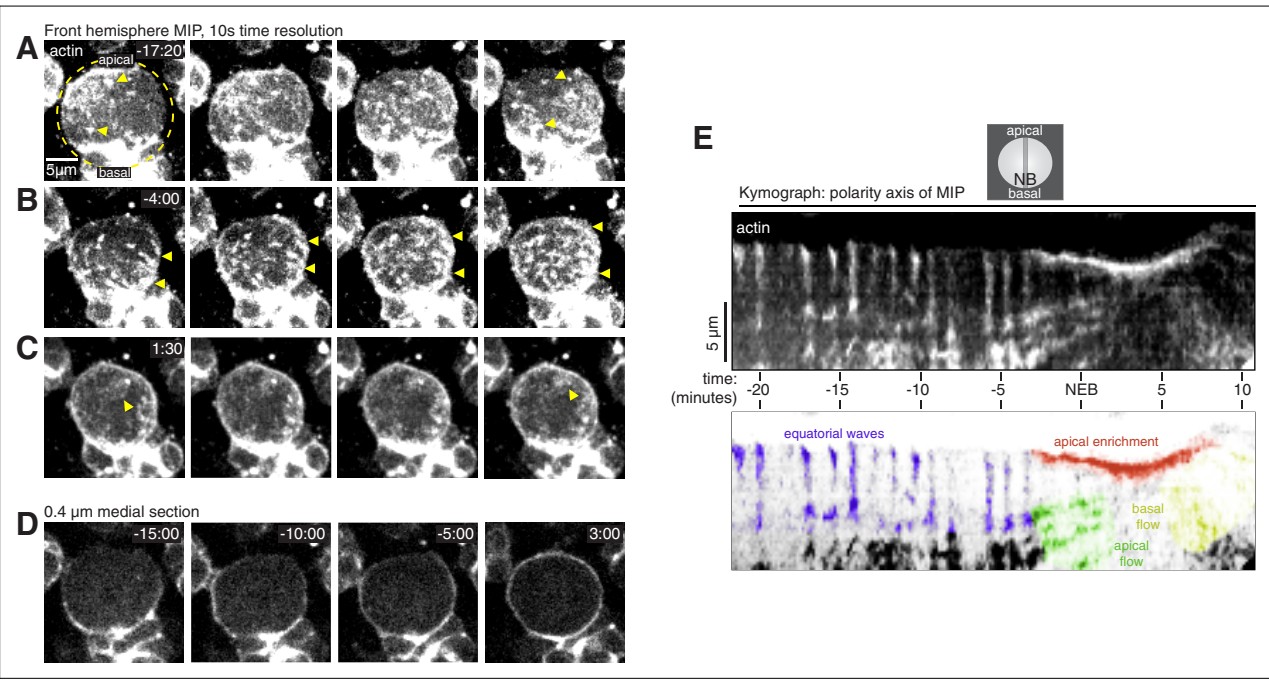

**Figure 1.** Cortical F-actin dynamics in asymmetrically dividing *Drosophila* larval brain neuroblasts. (**A**) Selected frames from *Video 1* showing cortical actin pulses during interphase. mRuby-LifeAct expressed via insc-GAL4/UAS ("actin") is shown via a maximum intensity projection (MIP) constructed from optical sections through the front hemisphere of the cell. The outline of the neuroblast is shown by a dashed yellow circle. In this example, the pulse moves from the upper left of the cell to the lower right. Arrowheads mark several cortical actin patches. Time (mm:ss) is relative to nuclear envelope breakdown. (**B**) Selected frames from *Video 1* as in panel A showing cortical actin moving apically. Arrowheads delineate apical and basal extent of dynamic actin. (**C**) Selected frames from *Video 1* as in panel A showing cortical actin enriched on the apical cortex. (**D**) Selected frames from *Video 1* showing how actin becomes cortically enriched near nuclear envelope breakdown (NEB). A single 0.4 µM medial section of the cortical actin signal is relatively discontinuous before NEB, with areas of very low actin signal, but becomes more evenly distributed as the cell rounds in mitosis (3:00 time point). (**E**) Kymograph constructed from frames of *Video 1* using sections along the apical-basal axis as indicated (NB, neuroblast). A legend with the features in the kymograph depicting the cortical dynamic phases and apical enrichment of actin is included below.

The online version of this article includes the following figure supplement(s) for figure 1:

**Figure supplement 1.** Imaging and analysis scheme for rapid, full volume imaging of *Drosophila* neuroblasts from larval brain explants.

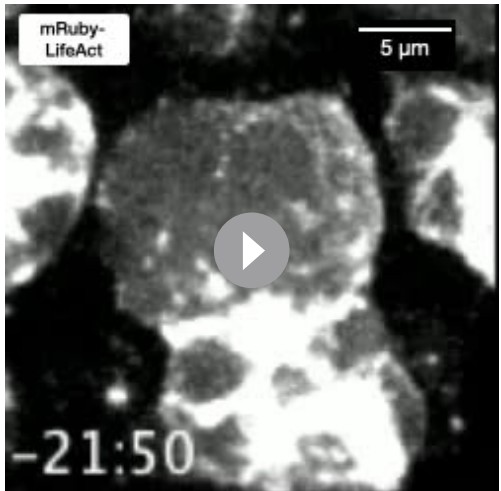

**Video 1.** Actin dynamics in a larval brain neuroblast. The mRuby-Lifeact sensor expressed from the UAS promoter and *insc-GAL4* (expressed in neuroblasts and their progeny) is shown with a maximum intensity projection of the front hemisphere of the cell.
https://elifesciences.org/articles/66574/figures#video1

## Results and discussion
### Cortical actin dynamics during the neuroblast polarity cycle

We imaged larval brain neuroblasts expressing an mRuby fusion of the actin sensor LifeAct (mRuby-LA) using spinning disk confocal microscopy. The localization of this sensor in neuroblasts has been reported (*Abeysundara et al., 2018*; *Roubinet et al., 2017*), but only during late mitosis. To follow cortical actin dynamics across full asymmetric division cycles, we collected optical sections through the entire neuroblast volume (~40 0.5 μm sections) at 10 s intervals beginning in interphase and through at least one mitosis (*Figure 1—figure supplement 1*). Maximum intensity projections constructed from these data revealed localized actin enrichments on the cortex, some of which were highly dynamic (*Figure 1* and *Video 1*). We observed three discrete phases of cortical actin dynamics that preceded the previously characterized basally directed flows that occur in late anaphase (*Roubinet et al., 2017*).

The interphase neuroblast cortex was a mixture of patches of concentrated actin, highly dynamic pulsatile waves that traveled across the entire width of the cell, and areas with little to no detectable actin (*Figure 1* and *Video 1*). Pulsatile movements consisted of irregular patches of actin forming on the cortex and rapidly moving across the surface before disappearing (*Figure 1A and E*). Concentrated actin patches were relatively static, but sometimes changed size over the course of several minutes. Static patches were mostly unaffected by the pulsatile waves that occasionally passed over them (*Figure 1A*). Pulses were sporadic in early interphase but became more regular near mitosis, with a new pulse appearing immediately following the completion of the prior one (*Figure 1E* and *Video 1*). The direction of the pulses during interphase was highly variable, but often along the cell's equator (i.e. orthogonal to the polarity/division axis). In general, actin in the interphase cortex was highly discontinuous and included large areas with little to no detectable actin in addition to the patches and dynamic pulses described above (*Figure 1D* and *Video 1*). Interphase pulses were correlated with cellular scale morphological deformations in which these areas of low actin signal were displaced away from the cell center while the cortex containing the actin pulse was compressed toward the center of the cell (*Figure 1D* and *Video 1*).

Near nuclear envelope breakdown (NEB), the static cortical actin patches began disappearing from the cortex while dynamic cortical actin reoriented toward the apical pole (*Figure 1* and *Video 1*). In contrast to the sporadic and relatively unoriented interphase pulses observed earlier in the cell cycle, the apically directed cortical actin dynamics that began near NEB were highly regular and were apically-directed (*Figure 1E* and *Video 1*). This phase of cortical actin dynamics continued until anaphase–consistent with previous descriptions of actin accumulation at the apical cortex throughout metaphase (*Barros et al., 2003*; *Tsankova et al., 2017*). Additionally, while the interphase cortex had areas with very little actin, actin was more evenly-distributed following the transition, as was apparent in medial sections (e.g. comparing –15:00 and 3:00 in *Figure 1D* and *Video 1*).

Another transition in cortical actin dynamics occurred shortly after anaphase onset when the apically directed cortical actin movements rapidly reversed direction such that the F-actin that had accumulated in the apical hemisphere began to move basally toward the emerging cleavage furrow (*Figure 1E* and *Video 1*). The basally directed phase of movement that begins shortly after anaphase onset and includes both actin and myosin II was reported previously (*Barros et al., 2003*; *Roubinet et al., 2017*; *Tsankova et al., 2017*).

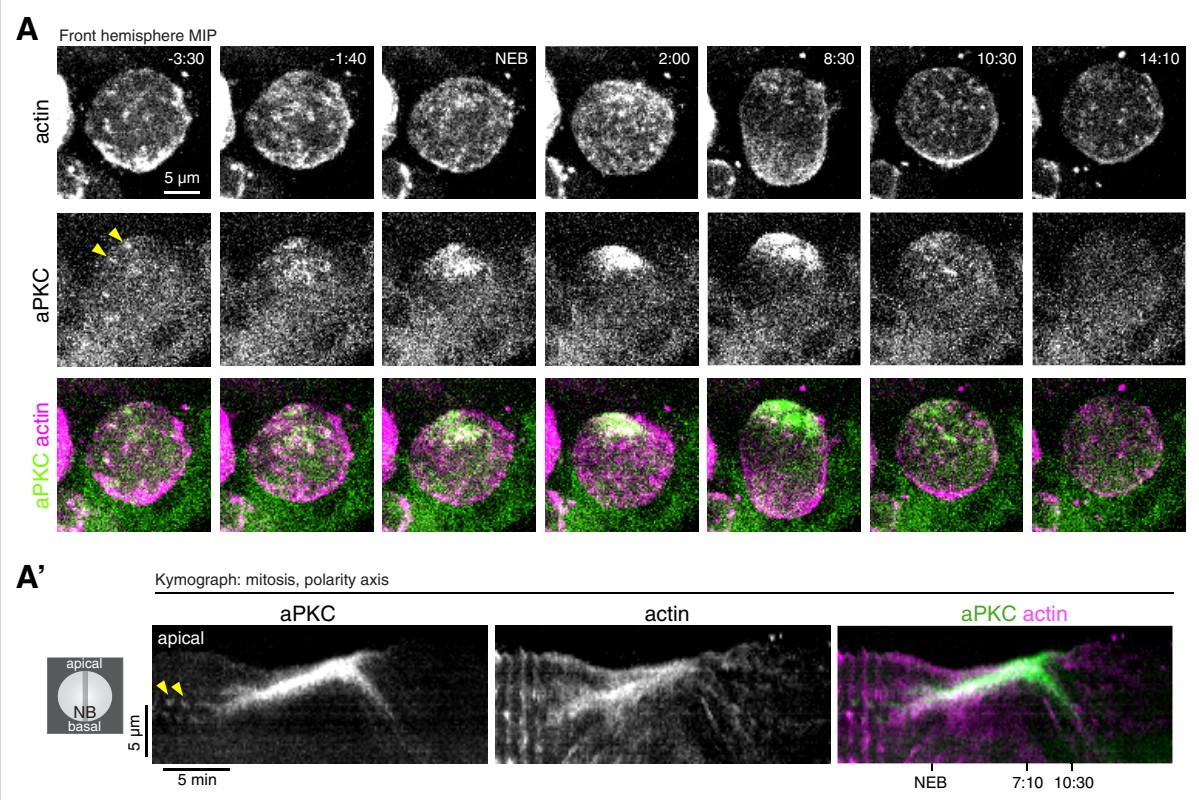

**Figure 2.** Coordinated actin and aPKC dynamics during the neuroblast polarity cycle.

The online version of this article includes the following video for figure 2:

**Figure 2—video 1.** Videos used to calculate when apically directed aPKC and actin movements begin.

https://elifesciences.org/articles/66574/figures#fig2video1

## Cortical actin and aPKC dynamics are coupled

Previously, we showed that Par polarity proteins undergo complex cortical dynamics during neuroblast asymmetric cell division and that polarity cycle movements require an intact actin cytoskeleton (*Oon and Prehoda, 2019*). Here, we have found that the cortical actin cytoskeleton is also highly dynamic at points in the cell cycle when Par proteins undergo coordinated cortical movement (*Figure 1* and *Video 1*). Furthermore, the transitions in cortical actin dynamic phases appeared to occur when similar transitions take place in the polarity cycle. We examined the extent to which cortical actin and aPKC dynamics are correlated by simultaneously imaging GFP-aPKC expressed from its endogenous promoter with mRuby-Lifeact (*Figure 2* and *Video 2*). Apical targeting of aPKC began approximately ten minutes before NEB, when small, discontinuous aPKC foci began to appear, as previously reported (*Oon and Prehoda, 2019*). The interphase pulses of actin had no noticeable effect on these aPKC enrichments, suggesting that at this stage of the cell cycle, cortical actin dynamics are not coupled to aPKC movement (*Figure 1C*).

While cortical actin and aPKC did not appear to be coupled during interphase, the two protein's movements were highly correlated beginning in early mitosis (*Figure 2* and *Video 2*). When

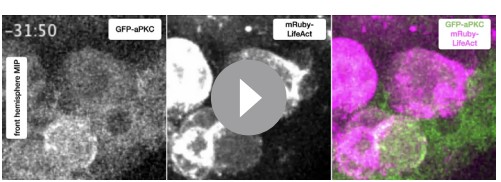

**Video 2.** Correlated dynamics of the Par protein aPKC and actin in a larval brain neuroblast. GFP-aPKC expressed from its endogenous promoter and the mRuby-Lifeact sensor expressed from the UAS promoter and *insc-GAL4* (drives expression in neuroblasts and progeny) are shown from simultaneously acquired optical sections with a maximum intensity projection of the front hemisphere of the cell.

https://elifesciences.org/articles/66574/figures#video2

cortical actin began flowing apically, the sparsely distributed aPKC patches that had accumulated on the cortex also began moving toward the apical pole. The transition to apically directed movement was nearly simultaneous for both proteins, although actin's apical movement began slightly before aPKC's (actin: 1.9 ± 1.0 min prior to NEB; aPKC: 1.3 ± 1.1 min; n = 13 neuroblasts with movies in *Figure 2—video 1*). Furthermore, while aPKC and actin both moved toward the apical cortex, actin dynamics occurred over the entire cortex whereas aPKC movements were limited to the apical hemisphere consistent with its specific targeting to this area (*Figure 2* and *Video 2*). The continuous apical movements resulted in the concentration of both aPKC and actin at the apical pole. Interestingly, however, once aPKC was collected near the pole into an apical cap, it appeared to be static while cortical actin continued flowing apically. This phase of dynamic, apically directed actin with an apparently static aPKC apical cap continued for several minutes (e.g. until approximately 7:10 in *Video 2*). At this point actin and aPKC movements reversed, moving simultaneously toward the basal pole and the emerging cleavage furrow (*Figure 2* and *Video 2*). We conclude that cortical actin and aPKC dynamics become highly correlated after interphase actin pulses transition to sustained, apically directed movements.

The correlation between actin and aPKC dynamics is consistent with our previous finding that depolymerization of actin with LatrunculinA (LatA) inhibits aPKC's apically directed cortical movements (*Oon and Prehoda, 2019*). We further examined the relationship between F-actin dynamics and aPKC using low doses of Cytochalasin D (CytoD) that inhibit actin dynamics but maintain cortical structure (*An et al., 2017*; *Mason et al., 2013*). The apically directed movements of actin during early mitosis were inhibited by CytoD with the cortex rapidly becoming relatively static (*Figure 3A and A'* and *Figure 3—video 1*). The loss of actin dynamics was immediately followed by cessation of apically directed aPKC movement such that it failed to form an apical cap (n = 11; neuroblasts shown in *Figure 3—video 2*). While both LatA and CytoD inhibited aPKC coalescence into an apical cap, we noticed that aPKC was maintained in the apical hemisphere for a longer period in CytoD-treated neuroblasts (*Figure 3B–D*, *Figure 3—video 3*, *Figure 3—video 4*, and neuroblasts used for measurements in *Figure 3—video 5*). Some localized enrichments remained in the apical hemisphere in neuroblasts treated with either drug, possibly due to their association with localized membrane enrichments, as recently reported (*LaFoya and Prehoda, 2021*). However, aPKC signal entered the basal hemisphere more rapidly in LatA- versus CytoD-treated neuroblasts (*Figure 3D*). Thus, aPKC appears to be better maintained in its polarized state when the cortical actin cytoskeleton remains intact (CytoD) compared to when it is completely depolymerized (LatA). We conclude that the cortical dynamics that drive the formation of the aPKC apical cap require the apically directed phase of actin dynamics that occurs during late prophase and metaphase. Furthermore, cortical actin structure may prolong the aPKC polarized state by slowing its diffusion into the basal hemisphere.

## Correlated phases of cortical myosin II and actin dynamics

The morphological changes in interphase cells (*Figure 1D* and *Video 1*) and cortical aPKC movements that were correlated with cortical actin dynamics in early mitosis (*Figure 2* and *Video 2*), are consistent with a force generating process. While actin can generate force directly through polymerization, contractile forces are generated by the combined activity of F-actin and myosin II (i.e. actomyosin), and cortical pulsatile contractions of actomyosin have been observed in many other systems (*Vicker, 2000*; *Munro et al., 2004*; *Michaux et al., 2018*). The localization of myosin II in neuroblasts has been described as uniformly cortical or cytoplasmic in interphase and before metaphase in mitosis (*Barros et al., 2003*; *Tsankova et al., 2017*; *Koe et al., 2018*). We used rapid imaging of the full cell volume, simultaneously following a GFP fusion of the myosin II regulatory light chain Spaghetti squash (GFP-Sqh) with mRuby-Lifeact, to determine if myosin II dynamics share any of the cortical dynamic phases we observed for actin. For each phase of actin dynamics, we found that myosin II is localized to the cortex in a similar pattern (*Figure 4* and *Video 3*), including during the apically directed continuous movements that polarize aPKC. Interestingly, however, the localization between the two was not absolute and there were often large cortical regions where the two did not colocalize in addition to the region where they overlapped (*Figure 4* and *Video 3*). A similar pattern of overlapping cortical actin and myosin II localization has been reported in the polarizing worm zygote (*Reymann et al., 2016*; *Michaux et al., 2018*). Given the similarities in behavior of actin and myosin II, we conclude that the

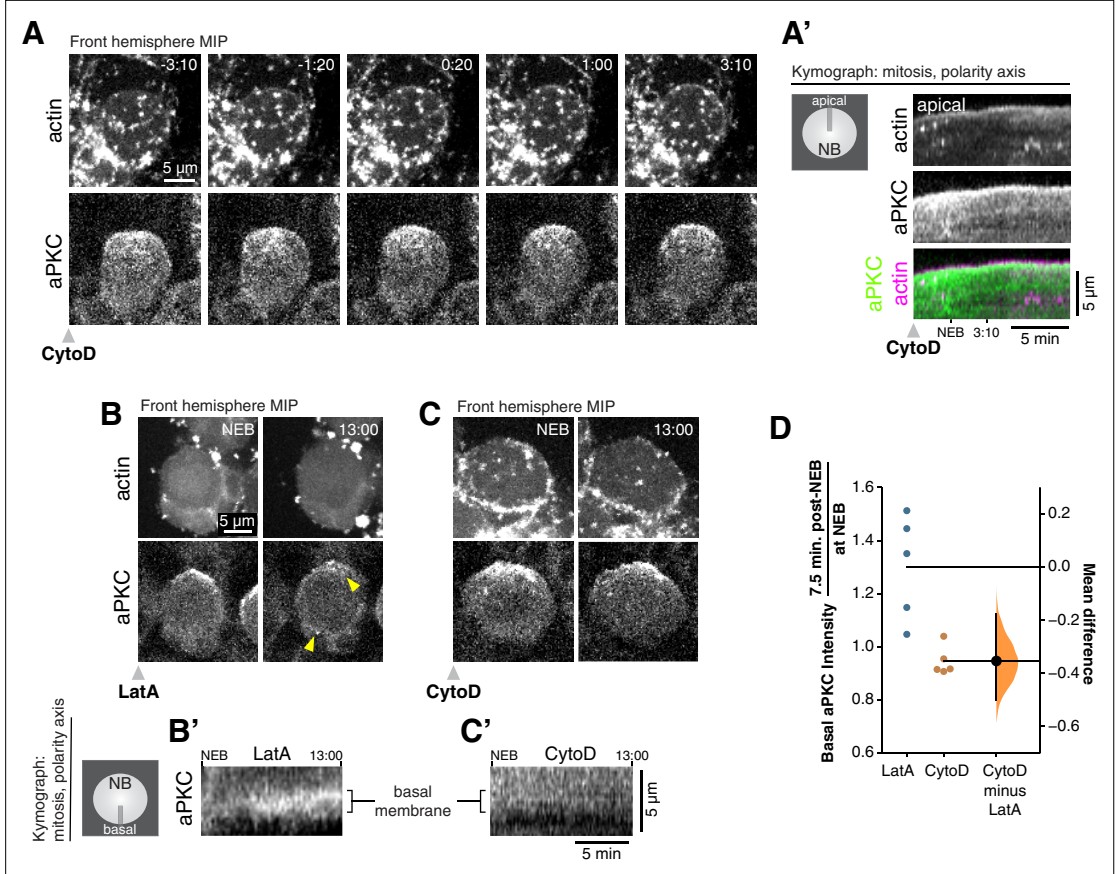

**Figure 3.** Cortical F-actin is required for aPKC coalescence and polarity maintenance. (**A**) Disruption of cortical F-actin dynamics using a low dosage (50 μM) of cytochalasin D (CytoD) causes immediate cessation of aPKC cortical movement. Selected frames from *Figure 3—video 1* showing the inhibition of actin dynamics and accompanying loss of apically-directed aPKC movements. aPKC-GFP expressed from its endogenous promoter ('aPKC') and mRuby-LifeAct expressed via insc-GAL4/UAS ('actin') are shown via a maximum intensity projection (MIP) constructed from optical sections through the front hemisphere of the cell. (**A'**) Kymograph made from *Figure 3—video 1* using a section of each frame along the apical-basal axis, as indicated. (**B**) Selected frames from *Figure 3—video 3* showing significant entry of aPKC into the basal hemisphere (i.e. depolarization) following LatA treatment. Upper arrowhead highlights a localized enrichment of aPKC that is retained in the apical region. Lower arrowhead highlights increased aPKC in basal hemisphere. (**B'**) Kymograph made from *Figure 3—video 3* using a section of each frame along the basal membrane near the cell's equator as indicated. (**C**) Selected frames from *Figure 3—video 4* showing the maintenance of aPKC polarization following CytoD treatment. (**C'**) Kymograph made from *Figure 3—video 4* using a section of each frame along the basal membrane as indicated. (**D**) Gardner-Altman estimation plot of the fold increase in basal aPKC membrane signal following Latrunculin A (LatA) and CytoD treatments. The ratio of aPKC membrane signal on the basal membrane shortly after NEB to that at NEB is shown for individual LatA- and CytoD-treated neuroblasts (shown in *Figure 3—video 5*), along with the difference in means of the measurements. Statistics: bootstrap 95 % confidence interval (bar in 'CytoD minus LatA' column shown with bootstrap resampling distribution).

The online version of this article includes the following video for figure 3:

**Figure 3—video 1.** Inhibition of cortical actin and aPKC dynamics by Cytochalasin D.
https://elifesciences.org/articles/66574/figures#fig3video1

**Figure 3—video 2.** Movies used to quantify apical movements of aPKC in Cytochalasin D-treated neuroblasts.
https://elifesciences.org/articles/66574/figures#fig3video2

**Figure 3—video 3.** Depolarization of aPKC induced by Latrunculin A.
https://elifesciences.org/articles/66574/figures#fig3video3

**Figure 3—video 4.** Maintenance of polarized aPKC in a Cytochalasin D-treated neuroblast.
https://elifesciences.org/articles/66574/figures#fig3video4

**Figure 3—video 5.** Movies used to quantify maintenance of aPKC in Latrunculin A- and Cytochalasin D-treated neuroblasts.
https://elifesciences.org/articles/66574/figures#fig3video5

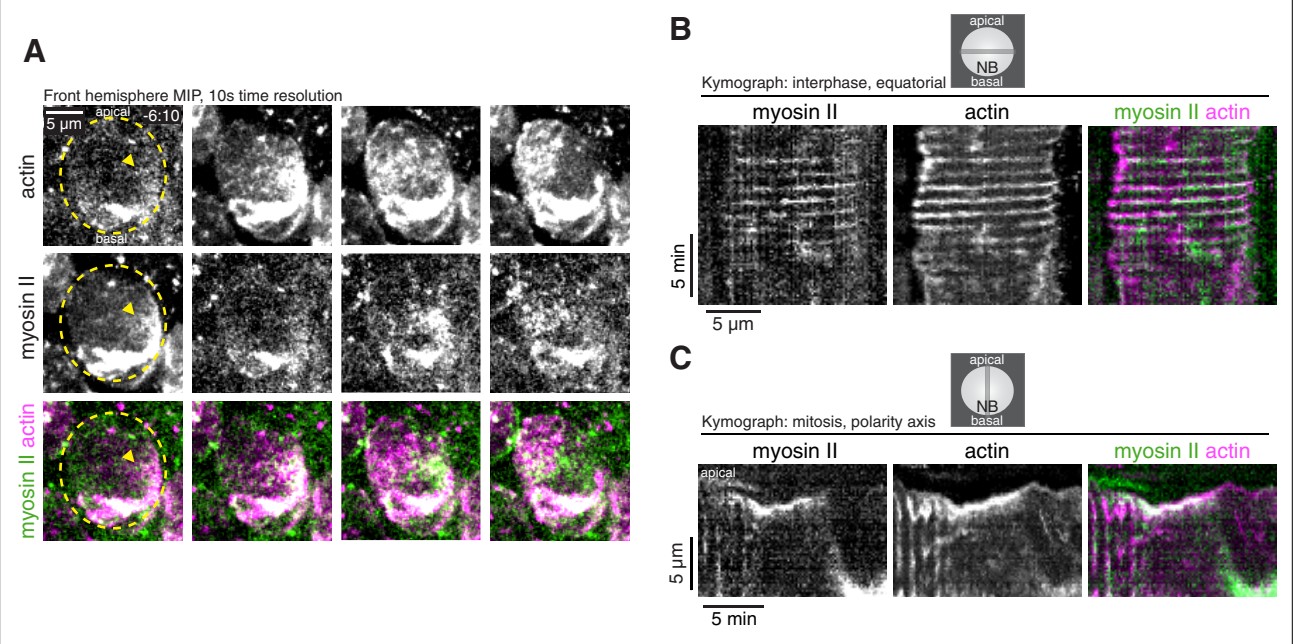

**Figure 4.** Dynamics of cortical actomyosin in asymmetrically dividing *Drosophila* larval brain neuroblasts. (**A**) Selected frames from *Video 3* showing cortical actomyosin dynamics. GFP-Sqh expressed from its endogenous promoter ('Myosin II') and mRuby-LifeAct expressed via worniu-GAL4/UAS ('actin') are shown via a maximum intensity projection (MIP) constructed from optical sections through the front hemisphere of the cell. The outline of the neuroblast is shown by a dashed yellow line and arrowheads indicate the starting position of the cortical patches. Time is relative to nuclear envelope breakdown. (**B**) Kymograph constructed from frames of *Video 3* during interphase using sections through the equatorial region of the cell as indicated. (**C**) Kymograph constructed from frames of *Video 3* during mitosis using sections along the polarity axis of the cell as indicated.

phases of cortical actin dynamics that occur during neuroblast asymmetric cell division include both actin and myosin II.

## Phases of cortical actomyosin dynamics coupled to neuroblast polarization, maintenance, and depolarization

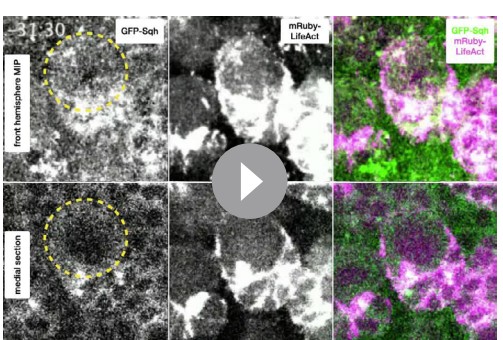

**Video 3.** Correlated dynamics of myosin II and actin in a larval brain neuroblast. GFP-Sqh (the myosin II regulatory light chain, Spaghetti squash) expressed from its endogenous promoter and the mRuby-Lifeact sensor expressed from the UAS promoter and *worniu-GAL4* (expressed in neuroblasts and their progeny) are shown from simultaneously acquired optical sections with a maximum intensity projection of the front hemisphere of the cell and the medial optical section. The neuroblast is highlighted by a dashed circle.
https://elifesciences.org/articles/66574/figures#video3

Our results reveal previously unrecognized phases of cortical actomyosin dynamics during neuroblast asymmetric division, several of which coincide with the neuroblast's cortical polarity cycle (*Figure 5*). During interphase, transient cortical patches of actomyosin undergo highly dynamic movements in which they rapidly traverse the cell cortex, predominantly along the cell's equator, before dissipating and beginning a new cycle (*Figure 1A*). Shortly after mitotic entry the movements become more continuous and aligned with the polarity axis (orthogonal to the equatorial interphase pulses). The transition to apically directed cortical actin movements occurs shortly before the establishment of apical Par polarity, when discrete cortical patches of aPKC undergo coordinated movements toward the apical pole to form an apical cap. Importantly, cortical actin dynamics are required for aPKC to coalesce into an apical cap (*Figure 3A* and *Figure 3—video 1*). Apically directed actin dynamics continue beyond metaphase when apical aPKC cap assembly is

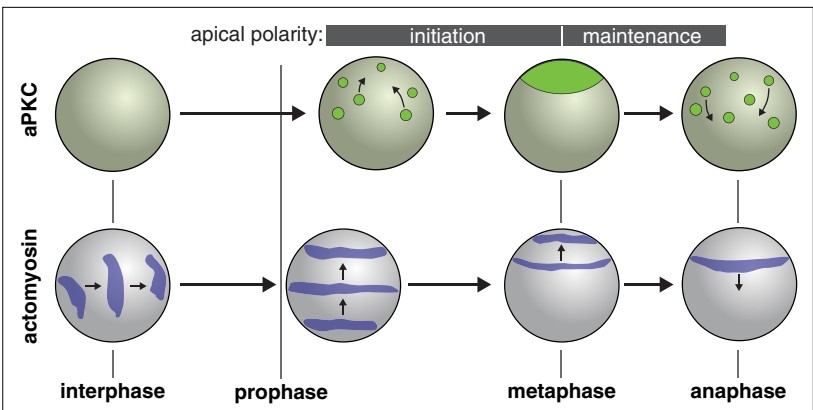

**Figure 5.** Model for role of actomyosin in neuroblast Par polarity. During interphase when aPKC is cytoplasmic, actomyosin pulsatile contractions are predominantly equatorial. During apical polarity initiation in prophase and shortly before when discrete aPKC cortical patches begin to coalesce, actomyosin transitions to more continuous movements directed toward the apical cortex. At anaphase apical actomyosin is cleared as it flows toward the cleavage furrow while the aPKC cap is disassembled.

completed (*Figure 2*), suggesting that actomyosin dynamics may also be involved in cap maintenance. A role for actomyosin in aPKC cap assembly and maintenance is supported by the lack of coalescence when the actin cytoskeleton is completely depolymerized (*Oon and Prehoda, 2019*), or when actin dynamics are inhibited but the cytoskeleton is left intact (*Figure 3B and B'*). The cycle of cortical acto-myosin dynamics is completed when the movement abruptly changes direction at anaphase leading to the cleavage furrow-directed flows that have been previously characterized (*Barros et al., 2003*; *Roubinet et al., 2017*). While we have examined the relationship between actomyosin dynamics and cortical protein polarity, we note that a neuroblast membrane polarity cycle was recently discovered and found to require the actin cytoskeleton (*LaFoya and Prehoda, 2021*). The mechanical phases of the membrane polarity cycle may be related to the phases of cortical actomyosin dynamics we report here.

While cortical actomyosin dynamics had not been reported during neuroblast polarization, myosin II pulses have been observed in delaminating neuroblasts from the *Drosophila* embryonic neuroecto-derm (*An et al., 2017*; *Simões et al., 2017*). The actomyosin dynamics reported here may be related to those that occur during delamination and provide a framework for understanding how actomyosin participates in neuroblast apical polarity. First, apically directed movements of actomyosin are consis-tent with the requirement for F-actin in the coalescence of discrete aPKC patches into an apical cap (*Figure 3*; *Oon and Prehoda, 2019*). How might cortical actomyosin dynamics induce aPKC coales-cence and maintenance? In the worm zygote, pulsatile contractions generate bulk cortical flows (i.e. advection) that lead to non-specific transport of cortically localized components (*Goehring et al., 2011*; *Illukkumbura et al., 2020*). Whether the cortical motions of polarity proteins that occur during the neuroblast polarity cycle are also driven by advection will require further study.

The more rapid depolarization of aPKC in Lat- compared to CytoD-treated neuroblasts (*Figure 3B–D*), is also consistent with a potentially passive role for the actin cytoskeleton in polarity maintenance. Complete loss of the cortical actin cytoskeleton (LatA; *Figure 3B*) leads to more rapid entry of aPKC into the basal neuroblast membrane compared to when cortical actin dynamics is inhibited but the structure maintained (CytoD; *Figure 3C*). The difference could arise simply from an increase in cortical diffusion constant when the cortical actin mesh is removed. In this case, the actin cytoskeleton would participate in Par polarity via at least two mechanisms: by generating non-diffusive movements of polarity proteins through actomyosin-generated cortical flows (*Figures 2 and 3*), and by maintaining the polarized state by slowing the rate of diffusion (*Figure 3*).

# Materials and methods

**Key resources table**

| Reagent type (species) or resource | Designation | Source or reference | Identifiers | Additional information |
|---|---|---|---|---|
| Genetic reagent (*Drosophila melanogaster*) | Lifeact-Ruby | Bloomington *Drosophila* Stock Center | BDSC:35545; FLYB:FBti0143328; RRID:BDSC_35545 | FlyBase symbol: P{UAS-Lifeact-Ruby}VIE-19A |
| Genetic reagent (*D. melanogaster*) | insc-Gal4 | Chris Doe Lab; Bloomington *Drosophila* Stock Center | BDSC:8751; FLYB:FBti0148948; RRID:BDSC_8751 | FlyBase symbol: P{GawB}insc$^{Mz1407}$ |
| Genetic reagent (*D. melanogaster*) | aPKC-GFP | François Schweisguth Lab; *Besson et al., 2015* | | BAC encoded aPKC-GFP |
| Genetic reagent (*D. melanogaster*) | wor-Gal4 | Chris Doe Lab; Bloomington *Drosophila* Stock Center | BDSC:56553; FLYB:FBti0161165; RRID:BDSC_56553 | FlyBase symbol: P{wor.GAL4.A}2 |
| Genetic reagent (*D. melanogaster*) | Sqh-GFP | *Royou et al., 2002* | | Expressed by natural *sqh* promoter |
| Chemical compound, drug | Latrunculin A | Sigma-Aldrich | Sigma-Aldrich: L5163 | (50 µM) |
| Chemical compound, drug | Cytochalasin D | Enzo Life Sciences | Enzo Life Sciences: BML-T109-0001 | (50 µM) |

## Fly strains and genetics

UAS-Lifeact-Ruby (Bloomington stock 35545), BAC-encoded aPKC-GFP (*Besson et al., 2015*) and Sqh-GFP (*Royou et al., 2002*) transgenes were used to assess F-actin, aPKC and myosin II dynamics, respectively. Expression of Lifeact was specifically driven in nerve cells upon crossing UAS-Lifeact-Ruby to *insc-Gal4* (1407-Gal4, Bloomington stock 8751) or to *worniu-Gal4* (Bloomington stock 56553). The following genotypes were examined through dual channel live imaging: BAC-aPKC-GFP / Y; insc-Gal4, +/+, UAS-Lifeact-Ruby; and; worGal4, Sqh-GFP, +/+, UAS-Lifeact-Ruby.

## Live imaging

Third instar larvae were incubated in 30 °C overnight (~12 hr) prior to imaging and were dissected to isolate the brain lobes and ventral nerve cord, which were placed in Schneider's Insect media (SIM). Larval brain explants were placed in lysine-coated 35 mm cover slip dishes (WPI) containing modified minimal hemolymph-like solution (HL3.1). Explants were imaged on a Nikon Ti2 microscope equipped with a Nikon 60 × 1.2 NA Plan Apo VC water immersion objective, a Yokogawa CSU-W1 spinning disk, and two Photometrics Prime BSI Scientific CMOS cameras for simultaneous dual channel imaging. Explants expressing Lifeact-Ruby, and either aPKC-GFP or Sqh-GFP were illuminated with 488 nm and 561 nm laser light. Approximately 40 optical sections with step size of 0.5 µm were acquired throughout the neuroblast volume at time intervals of 10–15 s. For drug treatments, the culture media surrounding the explants were brought to final concentrations of 50 µM LatA (2 % DMSO) or 50 µM CytoD (0.5 % DMSO) at the start of the imaging session.

## Image processing, analysis, and visualization

Movies were analyzed using the ImageJ (via the FIJI distribution) and Imaris (Bitplane) software packages. To quantify transitions between cortical actin dynamic phases, we identified the frame when actin or aPKC began moving toward the apical pole persistently (for at least several minutes, unlike the interphase, pulsatile motions). Likewise, the start of basally directed actin flow was indicated by the initial frame when apical actin or aPKC moved persistently toward the basal hemisphere. To investigate the effects of LatA and CytoD on neuroblast cortical dynamics, we examined aPKC and actin signals in early prophase to first determine if actin dynamics had ceased (i.e. the drug had taken effect), which typically occurred within five minutes of treatment for both drugs. To measure the degree to which aPKC polarity was maintained, we quantified the basal intensity of aPKC (via a 3-µm-thick line scan) across the basal membrane of a single medial section at NEB and at 7.5 min after NEB. The fold change in basal intensity across time was calculated using the following equation, with the mean background intensity obtained from a featureless area outside the neuroblast:

$$FoldChange \in BasalIntensity = \frac{(Meanbasalintensity - Meanbackgroundintensity)_{NEB+7.5min}}{(Meanbasalintensity - Meanbackgroundintensity)_{NEB}}$$

## Additional information

### Funding

| Funder | Grant reference number | Author |
|---|---|---|
| National Institutes of Health | GM127092 | Kenneth E Prehoda |

The funders had no role in study design, data collection and interpretation, or the decision to submit the work for publication.

### Author contributions

Chet Huan Oon, Conceptualization, Formal analysis, Investigation, Methodology, Visualization, Writing – original draft, Writing – review and editing; Kenneth E Prehoda, Conceptualization, Formal analysis, Funding acquisition, Methodology, Project administration, Supervision, Writing – original draft, Writing – review and editing

### Author ORCIDs

Kenneth E Prehoda (iD) http://orcid.org/0000-0003-4214-6158

### Decision letter and Author response

Decision letter https://doi.org/10.7554/eLife.66574.sa1
Author response https://doi.org/10.7554/eLife.66574.sa2

## Additional files

### Supplementary files

• Transparent reporting form

### Data availability

All data generated or analysed during this study are included in the manuscript and supporting files.

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
