## [Editor Report]

Oon and Prehoda report pulsatile contraction of apical membrane in the process of Par protein polarization in Drosophila neuroblasts. This explains how/why actin filament was required to localize/polarize Par complex. This very much resembles the observation in *C. elegans* embryos, and nicely unifies observations across systems.

---

## [Decision Letter]

**Decision letter after peer review:**

[Editors’ note: the authors submitted for reconsideration following the decision after peer review. What follows is the decision letter after the first round of review.]

Thank you for submitting your work entitled "Myosin II pulsatile contractions during the polarization of *Drosophila* neuroblasts" for consideration by *eLife*. Your article has been reviewed by 3 peer reviewers, including Yukiko M Yamashita as the Reviewing Editor and Reviewer #1, and the evaluation has been overseen by a Senior Editor.

Our decision has been reached after consultation between the reviewers. Based on these discussions and the individual reviews below, we regret to inform you that your work will not be considered further for publication in *eLife*. Although *eLife* encourages recent updates to previous work, our reviewers did not feel that the additional data constitutes the essential increment to justify publication.

This manuscript by Oon and Prehoda follows up on their previous work on how *Drosophila* neuroblasts may be polarized in preparation of asymmetric cell division. Now, by analyzing a probe for myosin, they show that neuroblasts undergo pulsatile contraction to achieve cell polarization. Although the reviewers appreciated the progress, they also felt that the study would require more in-depth analysis to warrant publication in *eLife*. Even for Research Advance format, of which contribution is expected to build upon a previous *eLife* paper, the reviewers felt that need of deeper, mechanistic analysis.

*Reviewer #1:*

This manuscript by Oon and Prehoda follows up on their previous work on how *Drosophila* neuroblasts may be polarized in preparation of asymmetric cell division. Now, by improving their live imaging conditions, they show that neuroblasts undergo pulsatile contraction to achieve cell polarization. This is reminiscent of *C. elegans* zygotes, unifying the known requirement of the same/similar set of genes between these two systems and the cell biological processes. Overall, this is a well-conceived, and well-executed work providing important insights into how Par system regulates cell polarity through the regulation of Myosin-dependent contractile system.

Their description about Myosin II's dynamic localization/ pulsatile movement is convincing. Unfortunately, due to photosensitivity, the were not able to image aPKC. I understand that this is a technical limitation difficult to overcome but drawing a conclusion about a relationship between Myosin II and aPKC purely based on previous studies, which had missed interesting aspects of Myosin movement, seems to be go beyond what can/should be extrapolated. If simultaneous imaging is not possible, at least can they conduct live imaging of aPKC only? Or any other 'apical' proteins (such as Baz, Par6) can be visualized simultaneously as Myosin II? Alternatively, now that their live imaging revealed the importance of imaging the full volume of the neuroblast, co-staining of fixed samples for aPKC and Myosin (combined with full volume imaging) might reveal additional spatial relationship between aPKC and Myosin during the phase of pulsatile movement? I feel that correlating Myosin II behavior with polarity complex is critically important for this manuscript to have a significant contribution to the field.

*Reviewer #2:*

In this manuscript, Oon and Prehoda used live-imaging technique to describe pulsatile dynamics of cortical myosin II during neuroblast asymmetric division and its linkage to the initiation and maintenance of apical Par polarity. While the dynamics described is novel to the field, the evidence provided on its causative relationship to the initiation and maintenance of apical Par polarity is weak and lacks convincing proofs. The reviewer suggests more experimental efforts be directed at strengthening the afore mentioned causative relationship.

In conclusion, it is in the view of the reviewer that the results in the manuscript are descriptive in nature and lacks mechanistic insights. Given the amount of the experiments required, the reviewer would not suggest that the manuscript be accepted for further evaluation.

List of suggested experiments:

– While the timing of apical cortical recruitment of aPKC has been described by the authors in their previous paper (Oon CH & Prehoda KE, *eLife* 2019) and analyzed in supplementary Figure 1D, it was under the wild-type condition, i.e. no GFP-Sqh overexpression. It will be prudent to revisit the aPKC recruitment dynamics under the condition of Sqh overexpression as the aPKC recruitment dynamics might differ.

– The causative relationship between myosin II and initiation and maintenance of apical polarity can be further strengthened via:

– Drug (i.e. blebbistatin, a myosin II inhibitor) treatment experiment to test whether the loss of myosin II activity will lead to delayed or distorted initiation and maintenance of apical polarity.

– Experiments using non-phosphorylatable and constitutive phosphorylated mutants of sqh, i.e. sqhA20A21 and sqhE20E21, respectively, to test whether these will lead to defect in initiation and maintenance of apical polarity.

– Test whether over-activating and under-activating myosin II activity via overexpressing myosin light chain kinase (MLCK) and myosin light chain phosphatase (MLCP) would affect apical polarity.

– Test for null allele of sqh, i.e. sqhAX3, in MARCM clones and investigate its effect on aPKC recruitment dynamics.

– Test for the role of upstream regulator of sqh, i.e. ROCK, in its downstream effect on aPKC recruitment dynamics.

– Test for drug inhibitor of ROCK, i.e. Y27632 compound, in is effect on aPKC recruitment dynamics.

– Test for under aPKC RNAi and mutant background whether GFP-Sqh dynamics is still the same, to further strengthen the argument the causative effect between GFP-Sqh and aPKC dynamics.

*Reviewer #3:*

This paper describes the high-resolution, live behavior of myosin-GFP as a *Drosophila* neuroblast undergoes asymmetric division. The data are analyzed in several ways for a detailed, high-quality description of myosin localization and dynamics during the polarization process. Because this is the only new data presented in the paper, the paper is very limited in scope. The data are also not surprisingly given the authors 2019 *eLife* publication about actin dynamics in this system, and other related papers. The paper is descriptive and confirmatory, and would be better suited for PLoS One or microPublication.

[Editors’ note: further revisions were suggested prior to acceptance, as described below.]

Thank you for submitting your article "Pulsatile actomyosin contractions underlie Par polarity during the neuroblast polarity cycle" for consideration by *eLife*. Your article has been reviewed by 3 peer reviewers, including Yukiko M Yamashita as the Reviewing Editor and Reviewer #1, and the evaluation has been overseen by Utpal Banerjee as the Senior Editor.

Essential revisions:

– The pulsatile nature of broad F-actin networks is evident during interphase, but these pulsations substantially subside upon entry into mitosis, and at this stage an apically directed flow of F-actin is the main behavior evident. This transition from pulses to flow is evident in both the movies and the kymographs of the F-actin probe. However, the authors state that the pulsations continue at the onset of mitosis and as the apical cap of aPKC matures. It is unclear whether the apical flow of aPKC and F-actin is associated with small-scale defined F-actin pulses, or small-scale random fluctuations of F-actin. The F-actin flow alone is an informative finding. The authors should consider revising their descriptions of these data (including in the manuscript title), or provide clearer examples of defined F-actin pulsations during the stage when aPKC polarizes.

– It would strengthen the paper considerably if the authors can show the causal link between actomyosin pulses and Par polarity by other experiments, such as the use of low dose cytochalasin D treatment, as reported in Mason et al., Nature Cell Biology 2014 and An Y. et al., Development, 2017.

– There are two papers that described pulsatile behavior during apical constriction in the process of neuroblast delamination in embryonic development. Although these are not the same process as what is described in this paper, it is likely that some underlying mechanisms are shared, and mention and reasonable discussion on these findings in relation to the current manuscript must be included.

An Y, Xue G, Shaobo Y, Mingxi D, Zhou X, Yu W, Ishibashi T, Zhang L, Yan Y. Apical constriction is driven by a pulsatile apical myosin network in delaminating *Drosophila* neuroblasts. Development. 2017 Jun 15;144(12):2153-2164. doi: 10.1242/dev.150763. Epub 2017 May 15. PMID: 28506995.

Simões S, Oh Y, Wang MFZ, Fernandez-Gonzalez R, Tepass U. Myosin II promotes the anisotropic loss of the apical domain during *Drosophila* neuroblast ingression. J Cell Biol. 2017 May 1;216(5):1387-1404. doi: 10.1083/jcb.201608038. Epub 2017 Mar 31. PMID: 28363972; PMCID: PMC5412560.

– In Figure 3A, following LatA treatment, apical aPKC foci/patch could still be observed at 19:40, while in Figure 2, aPKC starts to depolarize at 10:30 (mm:ss) after NEB. This seems to suggest that LatA treatment delays the depolarization of aPKC, which contradicts the conclusion that "aPKC is recruited to the apical cortex but rapidly depolarizes (Figure 3A and A')". Similarly, aPKC patch was still seen at 15:00 (Figure 3C), whereas apical aPKC should be completely gone in wild type condition at this time point (Figure 1). Also, in line 134-136, "we see aPKC begins diffusing away from the apical immediately following the disappearance of the cortical actin signal (Figure 3C,C' and Figure 3-video 3)" is an overstatement. It is more appropriate to conclude that aPKC had a slower diffusion rate following actin disruption.

– No listings of sample sizes in the main text, methods, figures and figure legends, which must be included.

– On page 4, line 105-122 (corresponding to Figure 2), authors described their observations at various time points such as 0:30 (line 109), -2:30 (line 112), 3:50 (line 115), and 7:20 (line 117), however, there are no corresponding still images for these time-points in Figure 2. Please show them.

– Lack of labeling to point out where readers should focus on in Figure 1B makes reader difficult to understand.

– Lack of abbreviation of nuclear envelope breakdown, NEB, in the main text (line 84-85).

– Lack of labeling for concentrated actin patches in Figure 1A.

– It would be better for readers if NEB is labeled in Figure 3A', 3B', and 3C'.

– Lack of sufficient description on the difference of LatA treatment in this study and previous study they published in *eLife* (2019).

*Reviewer #1:*

Oon and Prehoda report pulsatile contraction of apical membrane in the process of Par protein polarization in *Drosophila* neuroblasts. This explains how/why actin filament was required to localize/polarize Par complex. Specifically, using spinning disc confocal microscopy with high temporal resolution, they found the directed actin movement toward the apical pole, which nicely correlates with concentration of aPKC. They also show that myosin II is involved in this pulsatile movement of actin filament. This very much resembles the observation in *C. elegans* embryos, and nicely unifies observations across systems. Although descriptive in nature, I think this is an important observation and indicates a universal mechanism by which cells are polarized. I think this is a well-executed study and warrants publication in *eLife* as research advance.

*Reviewer #2:*

Previously, Oon and Prehoda showed apically directed movement of aPKC clusters during polarization of the neuroblast prior to asymmetric cell division. They found that these movements required F-actin, but the distribution of F-actin has only been reported for later stages of neuroblast polarization and division. Here, the authors report pulses of cortical F-actin during interphase, followed by an apically directed flow at the onset of mitosis, a strong apical accumulation of F-actin at metaphase and anaphase, followed by fragmentation and basally directed flow of the fragments. aPKC clusters are shown to colocalize with the F-actin networks as they flow apically. The F-actin networks are also shown have partial colocalization with non-muscle myosin II, suggesting a possible mechanism for their movement. Finally, the authors solidify the results of actin inhibitor studies from their 2019 study by showing that reported effects on aPKC localization are preceded by F-actin loss as would be expected but was not previously shown. Overall, the Research Advance extends the past study by more directly showing the involvement of F-actin and myosin in the apical localization mechanism of aPKC, and by describing F-actin and myosin dynamics prior to this transition. The following concerns should be addressed.

1. The pulsatile nature of broad F-actin networks is evident during interphase, but these pulsations substantially subside upon entry into mitosis, and at this stage an apically directed flow of F-actin is the main behavior evident. This transition from pulses to flow is evident in both the movies and the kymographs of the F-actin probe. However, the authors state that the pulsations continue at the onset of mitosis and as the apical cap of aPKC matures. It is unclear whether the apical flow of aPKC and F-actin is associated with small-scale defined F-actin pulses, or small-scale random fluctuations of F-actin. The F-actin flow alone is an informative finding. The authors should consider revising their descriptions of these data (including in the manuscript title), or provide clearer examples of defined F-actin pulsations during the stage when aPKC polarizes.

2. I checked the main text, methods, figures and figure legends, but could not find listings of sample sizes. Thus, the reproducibility of the findings has not been reported.

*Reviewer #3:*

In this revised manuscript (Oon and Prehoda), the authors performed additional live-imaging experiments and recorded aPKC and actin dynamics simultaneously in larval neuroblasts. They also provide evidence that aPKC polarization is lost upon F-actin disruption by Latrunculin A treatment. These are great improvements. The pulsatile dynamics of actin and myosin II showed in the manuscript are compelling. Images presented in this manuscript are of high-quality and impressive.

However, the pulsatile apical myosin network in delaminating neuroblasts in *Drosophila* embryos was reported previously (An Y. et al., Development, 2017). This important and relevant paper should be cited in the introduction of the current manuscript. Therefore, the finding on the pulsatile actomyosin in larval brain neuroblasts reported in this manuscript is not a total novel discovery. Another major concern is that Lat-A did not specifically disrupt actomyosin pulsatile movements, as it generally disrupts the F-actin network. So these experiments only strengthened the link between the F-actin network and Par polarity (which was already demonstrated in Kono et al., 2019; Oon 22 and Prehoda, 2019). Low doses of Cytochalasin D are known to disrupt myosin pulses still allowing the assembly of the actomyosin network (Mason et al., Nature Cell Biology 2014). The author should treat neuroblasts with low doses of CytoD to only disrupt actomyosin pulses, not the entire F-actin network, and examine the effect on Par polarity. It is also worthwhile to knockdown sqh to disrupt apical pulsatile actin dynamics. Besides, most of the concerns previously raised by the reviewer were not addressed in the revised manuscript.

---

## [Author Response]

[Editors’ note: the authors resubmitted a revised version of the paper for consideration. What follows is the authors’ response to the first round of review.]

Reviewer #1:This manuscript by Oon and Prehoda follows up on their previous work on how *Drosophila* neuroblasts may be polarized in preparation of asymmetric cell division. Now, by improving their live imaging conditions, they show that neuroblasts undergo pulsatile contraction to achieve cell polarization. This is reminiscent of *C. elegans* zygotes, unifying the known requirement of the same/similar set of genes between these two systems and the cell biological processes. Overall, this is a well-conceived, and well-executed work providing important insights into how Par system regulates cell polarity through the regulation of Myosin-dependent contractile system.Their description about Myosin II's dynamic localization/ pulsatile movement is convincing. Unfortunately, due to photosensitivity, the were not able to image aPKC. I understand that this is a technical limitation difficult to overcome but drawing a conclusion about a relationship between Myosin II and aPKC purely based on previous studies, which had missed interesting aspects of Myosin movement, seems to be go beyond what can/should be extrapolated. If simultaneous imaging is not possible, at least can they conduct live imaging of aPKC only? Or any other 'apical' proteins (such as Baz, Par6) can be visualized simultaneously as Myosin II? Alternatively, now that their live imaging revealed the importance of imaging the full volume of the neuroblast, co-staining of fixed samples for aPKC and Myosin (combined with full volume imaging) might reveal additional spatial relationship between aPKC and Myosin during the phase of pulsatile movement? I feel that correlating Myosin II behavior with polarity complex is critically important for this manuscript to have a significant contribution to the field.

We thank the reviewer for the enthusiasm for our work and recognize that the data in the original submission was limited because of our inability to simultaneously image aPKC and the cytoskeleton. We have worked hard to overcome this limitation and the revised manuscript includes data (Video 2) in which aPKC and actin dynamics are followed simultaneously. As implied by the reviewer, these data represent the most compelling case for the correlated dynamics of Par proteins and the cytoskeleton. We hope the reviewer agrees that the revision has a more complete mechanistic connection to our original paper – the actin cytoskeleton is required for aPKC dynamics (both our original paper and the current Research Advance), and actin and Par dynamics are highly correlated (the current Research Advance).

Reviewer #2:In this manuscript, Oon and Prehoda used live-imaging technique to describe pulsatile dynamics of cortical myosin II during neuroblast asymmetric division and its linkage to the initiation and maintenance of apical Par polarity. While the dynamics described is novel to the field, the evidence provided on its causative relationship to the initiation and maintenance of apical Par polarity is weak and lacks convincing proofs. The reviewer suggests more experimental efforts be directed at strengthening the afore mentioned causative relationship.In conclusion, it is in the view of the reviewer that the results in the manuscript are descriptive in nature and lacks mechanistic insights. Given the amount of the experiments required, the reviewer would not suggest that the manuscript be accepted for further evaluation.List of suggested experiments:– While the timing of apical cortical recruitment of aPKC has been described by the authors in their previous paper (Oon CH & Prehoda KE, eLife 2019) and analyzed in supplementary Figure 1D, it was under the wild-type condition, i.e. no GFP-Sqh overexpression. It will be prudent to revisit the aPKC recruitment dynamics under the condition of Sqh overexpression as the aPKC recruitment dynamics might differ.– The causative relationship between myosin II and initiation and maintenance of apical polarity can be further strengthened via:– Drug (i.e. blebbistatin, a myosin II inhibitor) treatment experiment to test whether the loss of myosin II activity will lead to delayed or distorted initiation and maintenance of apical polarity.– Experiments using non-phosphorylatable and constitutive phosphorylated mutants of sqh, i.e. sqhA20A21 and sqhE20E21, respectively, to test whether these will lead to defect in initiation and maintenance of apical polarity.– Test whether over-activating and under-activating myosin II activity via overexpressing myosin light chain kinase (MLCK) and myosin light chain phosphatase (MLCP) would affect apical polarity.– Test for null allele of sqh, i.e. sqhAX3, in MARCM clones and investigate its effect on aPKC recruitment dynamics.– Test for the role of upstream regulator of sqh, i.e. ROCK, in its downstream effect on aPKC recruitment dynamics.– Test for drug inhibitor of ROCK, i.e. Y27632 compound, in is effect on aPKC recruitment dynamics.– Test for under aPKC RNAi and mutant background whether GFP-Sqh dynamics is still the same, to further strengthen the argument the causative effect between GFP-Sqh and aPKC dynamics.

We have recast the paper to focus on actin dynamics rather than myosin for two reasons. First, a focus on actin is most relevant to our original paper (which this Research Advance builds on) as that paper showed that the actin cytoskeleton is required for aPKC dynamics. Second, using the Lifeact actin sensor allowed us to obtain simultaneous imaging for aPKC and the actin cytoskeleton (Video 2) which shows the highly correlated dynamics of the two. Thus, the mechanistic connection to the first paper is very clear – the actin cytoskeleton is required for aPKC dynamics (both our original paper and the current Research Advance), and actin and Par dynamics are highly correlated (the current Research Advance). The myosin II dynamics from the first version of this Research Advance are still included in this revised version, but only to show that myosin II is a component of the cortical pulses we discovered.

Reviewer #3:This paper describes the high-resolution, live behavior of myosin-GFP as a *Drosophila* neuroblast undergoes asymmetric division. The data are analyzed in several ways for a detailed, high-quality description of myosin localization and dynamics during the polarization process. Because this is the only new data presented in the paper, the paper is very limited in scope. The data are also not surprisingly given the authors 2019 eLife publication about actin dynamics in this system, and other related papers. The paper is descriptive and confirmatory, and would be better suited for PLoS One or microPublication.

*eLife* describes a Research Advance as “a short article that … build[s] on the original research paper in an important way.” By our estimation, our original submission satisfied this criteria but we recognize the reviewer’s objections. We have added significant new data to the revised paper most importantly including simultaneous imaging of aPKC and the actin cytoskeleton, clearly demonstrating their correlated dynamics. We believe this builds on our original paper, which had identified a requirement for the actin cytoskeleton in aPKC dynamics, by providing a role for actin in the polarity cycle.

The reviewer states that the data we report are “not surprising”. As the authors of the original paper, we were certainly surprised when we first observed pulsatile contractions of actomyosin in neuroblasts. For one thing, there are numerous reports of actin and myosin II dynamics in neuroblasts dating back to 2003 without any mention of these dynamics. Additionally, a very recent model for actomyosin’s role in neuroblast polarity (Hannaford et al. 2018 *eLife)* posits that it forms a static scaffold at the *basal* cortex. We respectfully disagree that the current neuroblast literature, including our 2019 paper, points clearly to the presence of actomyosin pulsatile contractions playing a role in apical polarization.

The reviewer states that the data we report are “descriptive and confirmatory”. While we do describe a previously unreported phenomenon, we disagree that this detracts from the impact of our work. This description provides mechanistic insight into the polarization process – we knew previously that the actin cytoskeleton was required but had no idea why. The current work fills this mechanistic gap. Since there is little to no explanation in the neuroblast literature for how the actin cytoskeleton contributes to apical polarity, we respectfully disagree that our results are confirmatory.

[Editors’ note: what follows is the authors’ response to the second round of review.]

Essential Revisions:– The pulsatile nature of broad F-actin networks is evident during interphase, but these pulsations substantially subside upon entry into mitosis, and at this stage an apically directed flow of F-actin is the main behavior evident. This transition from pulses to flow is evident in both the movies and the kymographs of the F-actin probe. However, the authors state that the pulsations continue at the onset of mitosis and as the apical cap of aPKC matures. It is unclear whether the apical flow of aPKC and F-actin is associated with small-scale defined F-actin pulses, or small-scale random fluctuations of F-actin. The F-actin flow alone is an informative finding. The authors should consider revising their descriptions of these data (including in the manuscript title), or provide clearer examples of defined F-actin pulsations during the stage when aPKC polarizes.The reviewers make an important point regarding the difference between actomyosin dynamics in interphase and mitotic neuroblasts. Whereas the interphase dynamics are more clearly pulsatile in the sense that they begin and end, the movements during mitosis appear to be more continuous. To account for this difference we have extensively revised our description throughout the manuscript (c.f. the paragraph beginning at line 84), and have also changed thetitle.– It would strengthen the paper considerably if the authors can show the causal link between actomyosin pulses and Par polarity by other experiments, such as the use of low dose cytochalasin D treatment, as reported in Mason et al., Nature Cell Biology 2014 and An Y. et al., Development, 2017.

We appreciate the suggestion to analyze the requirement for F-actin dynamics in aPKC coalescence in more detail and have added such an analysis to the revised manuscript. We determined the concentration of cytochalasin D that inhibited F-actin dynamics but did not ablate cortical F-actin (unlike our latrunculin A experiments in which cortical F-actin signal was undetectable following treatment). In neuroblasts treated with this concentration of the drug (50 µM CytoD), apically directed cortical flows of aPKC did not occur. In contrast to LatA treated neuroblasts, however, in which asymmetrically targeted aPKC rapidly depolarized (i.e. was not maintained), aPKC in cytoD-treated neuroblasts did not enter the basal hemisphere as rapidly (Figure 3D). These results support a model in which F-actin plays two roles in neuroblast polarization – an active role where actin dynamics are essential for apically directed cortical flows, and a more passive one in which the presence of cortical F-actin prevents depolarization (perhaps by slowing the rate of cortical diffusion).

– There are two papers that described pulsatile behavior during apical constriction in the process of neuroblast delamination in embryonic development. Although these are not the same process as what is described in this paper, it is likely that some underlying mechanisms are shared, and mention and reasonable discussion on these findings in relation to the current manuscript must be included.An Y, Xue G, Shaobo Y, Mingxi D, Zhou X, Yu W, Ishibashi T, Zhang L, Yan Y. Apical constriction is driven by a pulsatile apical myosin network in delaminating *Drosophila* neuroblasts. Development. 2017 Jun 15;144(12):2153-2164. doi: 10.1242/dev.150763. Epub 2017 May 15. PMID: 28506995.Simões S, Oh Y, Wang MFZ, Fernandez-Gonzalez R, Tepass U. Myosin II promotes the anisotropic loss of the apical domain during *Drosophila* neuroblast ingression. J Cell Biol. 2017 May 1;216(5):1387-1404. doi: 10.1083/jcb.201608038. Epub 2017 Mar 31. PMID: 28363972; PMCID: PMC5412560.

We agree with the reviewer that the processes described in these papers may be related to the phenomenon in our work and have gladly added a reference to them along with a short discussion of their possible relationship to our results.

– In Figure 3A, following LatA treatment, apical aPKC foci/patch could still be observed at 19:40, while in Figure 2, aPKC starts to depolarize at 10:30 (mm:ss) after NEB. This seems to suggest that LatA treatment delays the depolarization of aPKC, which contradicts the conclusion that "aPKC is recruited to the apical cortex but rapidly depolarizes (Figure 3A and A')". Similarly, aPKC patch was still seen at 15:00 (Figure 3C), whereas apical aPKC should be completely gone in wild type condition at this time point (Figure 1). Also, in line 134-136, "we see aPKC begins diffusing away from the apical immediately following the disappearance of the cortical actin signal (Figure 3C,C' and Figure 3-video 3)" is an overstatement. It is more appropriate to conclude that aPKC had a slower diffusion rate following actin disruption.

The reviewer is correct that the behavior of aPKC following LatA treatment is more nuanced than we had described in our original manuscript. While most aPKC does indeed rapidly dissipate upon LatA treatment, aPKC at localized enrichments or patches can persist. We have modified the text to explain this more clearly and included a quantification comparing the basal signal of aPKC shortly after treatment with LatA and CytoD. We have also included a citation to a recent paper from our lab suggesting that the LatA resistant regions are associated with polarized membrane domains.

– No listings of sample sizes in the main text, methods, figures and figure legends, which must be included.

We have included the sample size for each of the relevant experiments in the revised manuscript. Additionally, we have included the movies used for making measurements as figure videos (e.g. Figure 2-video 1 contains the 13 movies used to calculate when actin and aPKC began apical movement relative to NEB).

– On page 4, line 105-122 (corresponding to Figure 2), authors described their observations at various time points such as 0:30 (line 109), -2:30 (line 112), 3:50 (line 115), and 7:20 (line 117), however, there are no corresponding still images for these time-points in Figure 2. Please show them.– Lack of labeling to point out where readers should focus on in Figure 1B makes reader difficult to understand.– Lack of abbreviation of nuclear envelope breakdown, NEB, in the main text (line 84-85).– Lack of labeling for concentrated actin patches in Figure 1A.– It would be better for readers if NEB is labeled in Figure 3A', 3B', and 3C'.

We have remedied these items in the revised text and figures.

– Lack of sufficient description on the difference of LatA treatment in this study and previous study they published in eLife (2019).

We significantly changed this portion of the manuscript, adding the CytoD treatment to more specifically test the role of cortical actin dynamics in aPKC coalescence. The LatA-treated neuroblasts are only included in the revised manuscript to compare how quickly aPKC depolarizes relative to CytoD treatment.